# Age of Onset and Its Related Factors in Cocaine or Methamphetamine Use in Adults from the United States: Results from NHANES 2005–2018

**DOI:** 10.3390/ijerph182212259

**Published:** 2021-11-22

**Authors:** Alexandre Arthur Guerin, Jee Hyun Kim

**Affiliations:** 1Centre for Youth Mental Health, The University of Melbourne, Parkville, VIC 3052, Australia; 2Mental Health Theme, The Florey Institute of Neuroscience and Mental Health, Parkville, VIC 3052, Australia; 3IMPACT—The Institute for Mental and Physical Health and Clinical Translation, School of Medicine, Deakin University, Geelong, VIC 3220, Australia

**Keywords:** cocaine, methamphetamine, prevalence, frequency, onset, adolescence, risk factors, NHANES, epidemiology, addiction

## Abstract

Cocaine and methamphetamine are widely used illicit psychostimulants worldwide, with steadily increasing global markets that may impact on the frequency of use. Importantly, their use typically begins in youth. This is a particular concern because there is a link between the early age of first substance use and severity of substance use disorder later in life. The aim of the present study was therefore to investigate trends in prevalence, frequency, and age of onset of cocaine or methamphetamine use between 2005 and 2018 in the United States, using the nationally representative NHANES datasets. Factors associated with the ages of cocaine or methamphetamine use onset were also identified. From 2005 to 2018, prevalence and frequencies of cocaine or methamphetamine use increased, while age of onset remained relatively stable (~20 years of age). Annual household income, use of other substances, and intravenous drug use were identified as factors associated with early onset cocaine or methamphetamine use. These factors have important implications toward developing new prevention programs to reduce psychostimulant use.

## 1. Introduction

Psychostimulant use is a major global concern. It is estimated that 20 million people have used cocaine and 27 million people have used amphetamine-type stimulants (including methamphetamine) worldwide in the past year [1]. A recent report from the Substance Abuse and Mental Health Services Administration (SAMHSA) revealed that rates of cocaine and methamphetamine use may be growing in the United States (U.S.) [2].This may be due to increasing markets and psychostimulant supply [3]. Notably, the dangers associated with psychostimulant use are also on the rise in the U.S., with cocaine-related overdose deaths tripling, and methamphetamine-related deaths increasing fivefold between 2012 and 2018 [4].

Importantly, 0.7% of young people aged 12–17 have reported using cocaine in the past year, while 0.3% reported using methamphetamine in the U.S. [2]. This is a particular concern, as numerous studies have highlighted a link between the age of onset of substance use and severity of substance use disorder later in life [5,6,7,8,9,10,11]. In fact, evidence from a large dataset (*n* = 1567) revealed that people who start using stimulants in adolescence are 1.7 times more likely to develop clinical features of substance use disorders within two years following onset compared to people who started in adulthood [11]. Furthermore, a large twin study (*n* = 1733) showed that experimentation with drugs during adolescence can impact the development of substance use disorders later in life [12]. Another twin study reported that early onset substance use was associated with developing dependence, an association not explained by genetic or shared environmental factors [13]. More recently, a study found that people who started using substances earlier in life (age ≤ 17) were more likely than late-onset users (age > 18) to use drugs often, have a diagnosis of substance use disorder, and relapse after abstinence [14]. These findings highlight the necessity to gain a better understanding of trends of age of onset of psychostimulant use to develop effective new prevention programs and interventions against psychostimulant use.

While some evidence has suggested an association with previous substance use, personality disorders, and social factors with early age of onset of general substance use [15,16,17,18], few studies have examined factors associated with the age of onset of psychostimulant use specifically. To this end, large publicly available datasets such as the National Health and Nutrition Examination Survey (NHANES) are vital in finding potential factors associated with early onset psychostimulant use. This survey examines a nationally representative sample of ~5000 persons each year, and includes demographic, socioeconomic, dietary, and health-related questions [19]. Notably, the NHANES records patterns of substance use, including age of onset of cocaine or methamphetamine use. It therefore provides a unique opportunity to investigate trends of psychostimulant use over time, and to examine risk factors associated with age of onset. For the purpose of this study, we focused on cocaine and methamphetamine, as they are the most widely abused psychostimulants [2].

Therefore, the aim of the present study was to use the NHANES dataset (survey cycles 2005-06 to 2017-18) to (1) investigate trends in prevalence and frequency of current cocaine and methamphetamine use from 2005 to 2018 in the U.S. adult population; (2) investigate trends in age of onset of cocaine and methamphetamine use; and (3) identify factors associated with age of onset of cocaine and methamphetamine use. Given the increase in global supply of cocaine and methamphetamine [3] and based on other nationally representative reports [2], we hypothesize that prevalence and frequencies of both psychostimulants have increased over time in the U.S. We also hypothesize that the age of onset of cocaine and methamphetamine use may have decreased inversely to the increase in use and availability [20]. The identification of factors associated with age of onset is a highly novel exploratory aim, and we focus on sociodemographic factors, other drug use, and intravenous drug use associations because these factors have previously been suggested to be associated with early use and severity of use for methamphetamine and other substances [21,22,23,24]. The novelty of this exploratory aim precludes specific hypotheses, but any information gained in this regard will be valuable to inform future studies.

## 2. Materials and Methods

### 2.1. Study Population

The NHANES is a survey conducted every two years by the National Center for Health Statistics [19], which selects a nationally representative sample of the U.S. population in each cycle. Approximately 5000 people are randomly selected each year to complete the cross-sectional survey, which includes demographic, socioeconomic, dietary, and health-related questions. The sample for the survey is selected to represent the U.S. population of all ages. In addition, NHANES over-samples persons 60 and older, African Americans, and Hispanics to produce reliable statistics of potentially under-represented groups.

In this study, we analyzed NHANES data from the 2005–2006 to 2017–2018 survey cycles. The datasets are publicly available and were retrieved from the Centers for Disease Control and Prevention’s website. Data extracted represent participants over the age of 18 (*n* = 70,190), for which NHANES provides substance use data.

### 2.2. NHANES Data Extracted

#### 2.2.1. Drug Use Questionnaire

The NHANES Drug Use Questionnaire focuses on lifetime and current use of cannabis, cocaine, heroin, and methamphetamine, as well as intravenous drug use [19]. For the present study aims, we extracted data on past-month use and age of onset of cocaine and methamphetamine use for participants over the age of 18. Within the participants reporting use of cocaine and methamphetamine, we also extracted data on regular cannabis use (cannabis use every month for a year), lifetime heroin, and lifetime intravenous drug use. Regular tobacco use (tobacco use most days in the past month) data were extracted from the Cigarette Use Questionnaire.

#### 2.2.2. Demographic Characteristics

The NHANES demographics questionnaire collects information on key demographics, including age, sex, ethnicity, education, marital status, military service status, country of birth, citizenship, years of U.S. residence, household and family income, and household composition. For the present study aims, we extracted the following variables of interest: age (in years), sex (male/female), education level (from below 9th grade to college graduate or above), and annual household income (from $0–$4999 to $100,000 and over).

### 2.3. Statistical Analyses

All analyses were performed in IBM SPSS Statistics 27 (IBM Corp., Armonk, NY, USA). The prevalence and frequencies of past-month cocaine and methamphetamine use were calculated and extrapolated to the U.S. adult population using the weights provided by NHANES. Weights are created to account for the complex survey design (including oversampling), survey non-response, and post-stratification adjustment to match total population counts from the U.S. Census Bureau. It is a measure of the number of people in the population represented by that sample person [19]. Sample weights are created in three steps; base weights for individuals are first computed, then adjusted for non-response, and finally post-stratified to match the population control totals for each sampling subdomain. The estimated prevalence is reported as a percentage of the U.S. adult population. Prevalence calculations were based on the U.S. population aged 18 years-old and above, provided by the U.S. Census Bureau. To examine the changes between each survey year, the percent change in prevalence was calculated for each cycle. The estimated frequencies were reported as the number of individuals reporting past-month use. The trends in age of onset of cocaine and methamphetamine use were reported as median age in years with interquartile range for each NHANES cycle, only for people who reported using cocaine and/or methamphetamine. Descriptive analyses were used to identify trends in prevalence, frequencies, and onset. In addition, linearity of trends was tested using simple linear regressions.

Hierarchical linear regressions were conducted to investigate factors associated with age of onset of cocaine and methamphetamine use, after controlling for cycle year, and age at screening. Control variables were entered in the first block of the regression. Variables of interest included in the analyses were demographic characteristics (sex, level of education, annual household income), use of other drugs (tobacco, cannabis, heroin, cocaine, or methamphetamine) and intravenous drug use, and were entered in the second block of the regression. The statistics of interest were the change in F and *p*-value associated with inclusion of variables in the model, and the β values of individual variables. Regressions were conducted by pooling all NHANES datasets available (2005–2018) using unique identification numbers for each variable. In total, data from 3975 people who use cocaine, and 1606 people who use methamphetamine were included in these analyses.

## 3. Results

### 3.1. Prevalence and Frequencies of Past-Month Cocaine or Methamphetamine Use

While past month use of both substances in the U.S. adult population has reached a new maximum in prevalence (Figure 1) and frequency (Table 1) in recent years, these changes were not statistically significant when tested for linear trends (ps > 0.05). The prevalence (and frequency) of past-month cocaine use increased from 1.178% (~2.22 million people) in 2005–2006 to 1.82% (~3.97 million people) in 2017–2018. Likewise, the prevalence (and frequency) of past-month methamphetamine use increased from 0.564% (~1.06 million people) in 2005–2006 to 0.796% (~1.74 million people) in 2017–2018. Notably, there was a marked increase in prevalence between the last two survey cycles, with a percent change increase of 57.1% and 97.5% from 2015–2016 to 2017–2018 for cocaine and methamphetamine, respectively. At all timepoints, past-month cocaine use was higher than methamphetamine use.

### 3.2. Trends in Age of Onset of Cocaine and Methamphetamine Use

Age of onset of cocaine use remained relatively stable from 2005–2018 in the U.S. population and did not follow a linear trend (ps < 0.05), with a median age of 20 for six of the seven NHANES cycles, and highest median age of onset of 21 years reported in 2011–2012 (Figure 2). Age of onset of methamphetamine use also remained stable in 2005–2018, ranging from 19 to 21 years of age.

### 3.3. Factors Associated with Age of Onset of Cocaine and Methamphetamine Use

To investigate factors associated with age of onset of cocaine and methamphetamine use, two hierarchical multiple regressions were conducted by pooling all NHANES datasets available (2005–2018). In total, data from 3975 people who use cocaine, and 1606 people who use methamphetamine were included in these analyses. It should be noted that the cocaine and methamphetamine participants were not mutually exclusive (i.e., a participant may have reported both lifetime cocaine and methamphetamine use). Variables of interest included demographic characteristics (sex, education, and annual income) and use of other drugs (tobacco, cannabis, heroin, cocaine, methamphetamine, intravenous use).

#### 3.3.1. Demographic and Other Drug Use Characteristics

Figure 3 and Table 2 describe the demographic and other drug use characteristics only of people reporting lifetime use of cocaine and methamphetamine in the NHANES 2005–2018. People in the cocaine and methamphetamine groups were predominantly male (62.7% and 63.9%, respectively), and predominantly were high school graduates or had some college experience (59.5% and 65.2%, respectively). There were no clear trends in annual household income. People who reported lifetime cocaine use also reported high rates of regular tobacco use (75.1%) and cannabis use (47%). People who reported lifetime methamphetamine use reported very high rates of lifetime cocaine (91.5%) and regular tobacco use (81.7%), and high rates of cannabis use (52.7%). People who used methamphetamine were more likely to have used drugs intravenously (20.7%) compared to people who used cocaine (11.9%).

#### 3.3.2. Age of Onset of Cocaine Use

The results of the hierarchical regression to examine associations with age of onset of cocaine use is presented in Table 3. Survey cycle year and age at screening were entered in Block 1. Survey cycle year and age at screening alone explained 9.4% of the variation in age of onset of cocaine use. Both were significantly associated with age of onset, with age of onset decreasing as time passed (β = −0.054, *p* = 0.015), and age at testing associated with later age of onset (β = 0.304, *p* < 0.001). Addition of demographic and drug use variables of interest in Block 2 significantly improved the predictive value of the model [FΔ(8, 1821) = 16.629, *p* < 0.001], and explained a further 6.2% of the variance. Notably, survey cycle year was no longer significant. The overall model statistically significantly predicted age of onset of cocaine use [F(10, 1821) = 33.643; *p* < 0.001; adjusted R^2^ = 0.151]. Annual household income was significantly associated with age of onset of cocaine use (β = −0.091; *p* < 0.001), with higher income associated with earlier age of onset. Other drug-use was also associated with age of onset of cocaine use. Regular tobacco use was associated with later cocaine use onset. Regular cannabis use (β = −0.095; *p* < 0.001), and lifetime methamphetamine (β = −0.134; *p* < 0.001) and intravenous use (β = −0.086; *p* = 0.001) were all associated with earlier age of onset. Sex, education, and lifetime heroin use were not associated with age of onset of cocaine use (ps > 0.05).

#### 3.3.3. Age of Onset of Methamphetamine Use

The results of the hierarchical regression to examine associations age of onset of methamphetamine use are presented in Table 4. Survey cycle year and age at screening were entered in Block 1. Survey cycle year and age at screening alone explained 8.9% of the variation in age of onset of methamphetamine use. Age at screening was significantly associated with age of onset of methamphetamine use (β = 0.291, *p* < 0.001), with age at screening associated with later age of onset. The addition of demographic and drug use variables of interest in Block 2 significantly improved the predictive value of the model [FΔ(8, 827) = 6.947, *p* < 0.001], and explained a further 5.7% of the variance. The overall model statistically significantly predicted age of onset of methamphetamine use [F(10, 827) = 14.196; *p* < 0.001; adjusted R^2^ = 0.136]. Annual household income was significantly associated with age of onset (β = −0.133; *p* < 0.001), with higher income associated with earlier onset. Regular tobacco use was associated with later methamphetamine use onset (β = 0.132; *p* < 0.001). Regular cannabis use (β = −0.090; *p* = 0.007) and lifetime intravenous use (β = −0.095; *p* = 0.012) were associated with age of onset, with both variables associated with earlier age of onset of methamphetamine. Sex, education and lifetime heroin and cocaine use were not associated with age of onset of methamphetamine use (ps > 0.05).

## 4. Discussion

In this study, we analyzed the publicly available NHANES datasets to investigate trends in prevalence, frequencies, and age of onset of cocaine and methamphetamine use in the U.S. from 2005 to 2018. We observed that past month use of both cocaine and methamphetamine have increased over time, with cocaine consistently used more than methamphetamine. In addition, we found that age of onset of cocaine or methamphetamine use remained relatively stable from 2005–2018. Hierarchical regression analyses revealed that higher annual household income was associated with earlier age of onset for participants that use cocaine and/or methamphetamine. In addition, cannabis, methamphetamine, and intravenous drug-use were associated with earlier age of onset of cocaine use, whereas only cannabis and intravenous drug-use were associated with earlier age of onset of methamphetamine use.

### 4.1. Cocaine and Methamphetamine Use Has Increased from 2005 to 2018

Overall, trends in prevalence and frequencies of recent cocaine or methamphetamine use were comparable to findings from other national samples, including the National Survey on Drug Use and Health (NSDUH; SAMHSA, 2020). It should however be noted that prevalence of cocaine use reported in the NSDUH was higher than the estimate in the present study. In their latest report, the NSDUH estimates that 5.5 million people (or 2% of the U.S. population) used cocaine in 2017–2018, whereas we reported an estimated 3.97 million people (1.8% of the adult population) based on the NHANES. Discrepancies may arise from the differences in survey design; past-year substance use was recorded in the NSDUH report, whereas we reported past-month (i.e., more recent) use. In addition, the NSDUH collects substance use data for people aged 12 and above, whereas the NHANES only records it for people aged 18 and above.

Nonetheless, data from both samples highlight an increase in recent cocaine and methamphetamine use from 2005 to 2017. This may be due to increasing psychostimulant availability worldwide [3]. In fact, seizure of methamphetamine is on the rise, with a 43% increase compared to the previous year. In addition, the largest quantities of methamphetamine seized in 2019 were seized in the U.S., with the quantity increasing eightfold between 2009 and 2019 [3]. Likewise, quantities of cocaine seized between 2009 and 2019 have increased by 90% globally, likely reflecting increased cocaine manufacturing and trafficking [3]. Interestingly, the present study showed that cocaine use in the U.S. temporarily dipped in the 2013-14 survey years, with a prevalence of 0.70% (~2.36 million people), compared to >1% in the previous and following cycle years. The dip in prevalence seems to be associated with an all-time low coca bush cultivation in 2014 [3], which likely impacted supply. Overall, data from the NHANES show that prevalence and frequencies of psychostimulant use have increased from 2005 to 2018, most likely due to increased manufacturing, trafficking, and availability of cocaine and methamphetamine around the globe. This trend is comparable to data presented in other nationally representative surveys. Importantly, the increase in cocaine and methamphetamine use appears to be largely supply driven. Findings from this study challenge the idea of the “War on Drugs” [25], that heavily involves incarceration of drug users. The present findings highlight the need for law enforcements to target suppliers and manufacturers, rather than users.

### 4.2. Factors Associated with Age of Onset of Cocaine or Methamphetamine Use

The median age of onset of cocaine or methamphetamine use remained stable between 2005 and 2018 in the U.S. in the present study (Figure 2). It ranged from 20 to 21 years for cocaine, and 19 to 21 years for methamphetamine. The stable trend is consistent with results from the NSDUH [26], and the Monitoring the Future national survey [27]. This is also consistent with figures from other countries; in Australia, the median age of onset of cocaine and methamphetamine use is 22 and 20 years, respectively [28]. In the European Union, mean age at first cocaine use is 23 years old, compared to 22 years for methamphetamine [29]. These median ages of onset of ~20 is a concern because the adolescent brain is still maturing well into our 20s [30,31]. It should also be noted that the result from this study is likely an overestimation of the actual age of onset, as all participants were aged 18 and above.

Results from the hierarchical regressions (Table 3 and Table 4) revealed that annual household income and use of other drugs were associated with age of onset of cocaine and methamphetamine, even after controlling for survey year and age at screening. Notably, we found that higher annual household income was associated with an earlier age of onset of both psychostimulants. This is consistent with a previous study showing an association between higher household income and higher probability of binge drinking, cannabis and cocaine use in early adulthood [21,32]. Results in the present study extend these findings to methamphetamine use. Demand for substances tend to be price sensitive [33,34], and it is therefore possible that substance use becomes more likely as income increases. Education and awareness of these findings in people from higher socioeconomic backgrounds in schools and family settings may therefore be a promising avenue to reduce early onset substance use. In addition, results from the present study may reduce stigma towards people who use psychostimulants, methamphetamine in particular. People who use substances experience a high level of stigma such as stereotyping (i.e., being perceived as dangerous, immoral, and having a lower socioeconomic status), discrimination and eliciting negative emotional reactions [35], which may hinder access to treatment and care [36,37]. Results from our study suggest that this population may in fact be high income earners. Conversely, because this study examined anyone who used methamphetamine in the past month (i.e., both people diagnosed with methamphetamine use disorder, and recreational users), it is possible that high-functioning, high-earning individuals skew the results.

Use of other drugs was also significantly associated with the age of onset of cocaine or methamphetamine use. Specifically, an earlier onset of cocaine use was associated with regular cannabis use and lifetime methamphetamine use, whereas only regular cannabis use was associated with earlier onset of methamphetamine use. A previous study showed that cannabis use may lead to initiation of cocaine early in life [16]. Results in the present paper suggest that cannabis use may also be a risk factor for early methamphetamine use. Considering that other studies have found that the average age of initiation of cannabis use is lower than cocaine or methamphetamine use [2], it is possible that initiation of cannabis use predates experimentation with psychostimulants [16].

We also identified an association between intravenous drug use and age of onset of both cocaine and methamphetamine use. Intravenous use is associated with higher drug bioavailability [38,39], which may lead to ongoing use of the drug into adulthood. People who use multiple substances early in life are more likely to inject drugs compared to people who start using later [40,41], which is consistent with the use of several classes of drugs reported in the previous paragraph. Conversely, it may be that an earlier age of onset of cocaine and/or methamphetamine use lead to experimentation with intravenous drug later in life. It is critical to note that due to the cross-sectional nature of the NHANES, the directionality of the associations between age of onset of psychostimulant use and these factors could not be determined. Nonetheless, it appears that early psychostimulant use is strongly associated with lifetime use of other substances. Using multiple substances may lead to poorer treatment outcomes, including lower retention rates, and higher rates of relapse [23,41,42]. Therefore, identifying people who are more at risk of using multiple substances in their lifetime is critical.

### 4.3. Limitations

While the use of a large, publicly available, and nationally representative dataset is a strength of this paper, the NHANES is a cross-sectional survey. That is, further longitudinal studies are required to ascertain the direction of associations between age of onset of psychostimulant use and factors identified in the present report. In addition, only drug use data for participants aged 18 and over were publicly available, which may have overestimated the ages of onset. Future studies should aim to examine risk factors in the population under 18. The NHANES Drug Use Questionnaire is a self-report questionnaire, and only captures current/lifetime use rather than formal diagnosis of a substance use disorder. While self-reported psychostimulant use reliably correlates with actual use [43,44], without biochemical verification it is impossible to ascertain whether past-month use was over- or under-reported. It would also be informative to investigate the association between age of onset of psychostimulant and odds of development a formal stimulant use disorder later in life. Lastly, evidence suggests that the length of recall period in self-reported health care questions varies between survey questions [45], and may affect the results of the survey. In the present study, participant age ranged from 18 to 69 years old. The age of onset measure in particular may have been subject of recall bias, as older participants may not correctly recall their exact age at first use. While we did control for the age at survey in our regressions, future studies should aim to reproduce these findings in a younger population with a narrower age range.

## 5. Conclusions

The present examination of the nationally representative NHANES datasets from 2005 to 2018 shows that prevalence and frequencies of both cocaine and methamphetamine use are on an upward trajectory, which is in concert with increased markets and availability worldwide. Age of onset of cocaine or methamphetamine use has remained relatively stable between 2005 and 2018 (~20 years of age). This median age of onset indicates that half of the participants initiated cocaine or methamphetamine use before 20 years of age, which is a significant concern as substance use during adolescence can lead to poorer outcomes later in life, and our brain continues to develop well into our 20s. Annual household income, use of other substances, and intravenous drug use were identified as factors associated with early onset cocaine or methamphetamine use. Taken together, these findings challenge the existing approach in the “War on Drugs”. Targeting supply rather than users may reduce prevalence. Considering the strong association between early onset of use and higher household income, education programs should target schools of higher sociodemographic status to prevent early psychostimulant use. In addition, this study suggests that the use of one drug early in life may lead to the use of other substances later. Promoting the treatment of people who started using drugs early with financial aids and subsidies may therefore present a unique opportunity to prevent their transition to other drug use and dependence.

## Figures and Tables

**Figure 1 ijerph-18-12259-f001:**
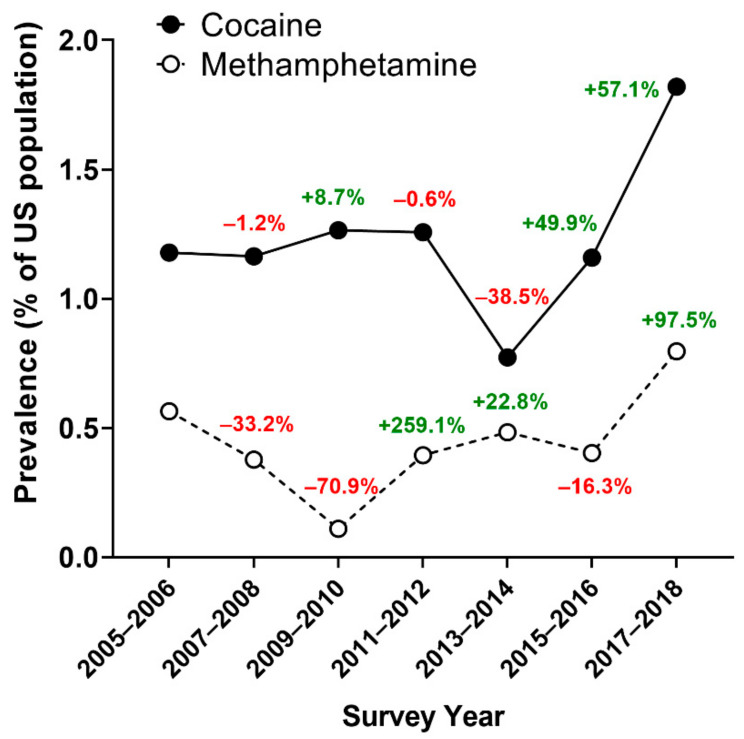
Estimated prevalence of past-month cocaine and methamphetamine use by survey years, NHANES 2005–2018. Prevalence is represented as the percentage of the U.S. population reporting using cocaine or methamphetamine in the past month. Values on the graphs represent percent change compared to the previous survey year, with values in green indicating an increase and values in red indicating a decrease compared to the previous cycle.

**Figure 2 ijerph-18-12259-f002:**
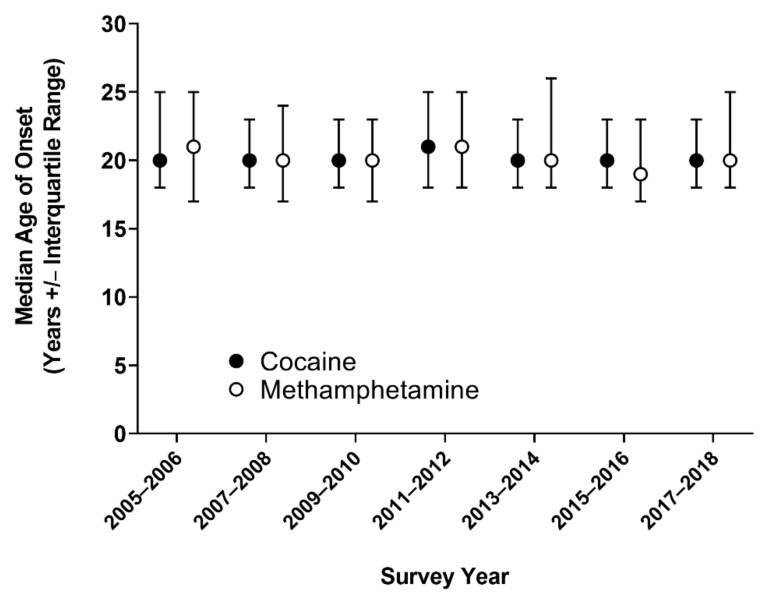
Median age (and interquartile range) of onset of cocaine and methamphetamine use by survey years, NHANES 2005–2018.

**Figure 3 ijerph-18-12259-f003:**
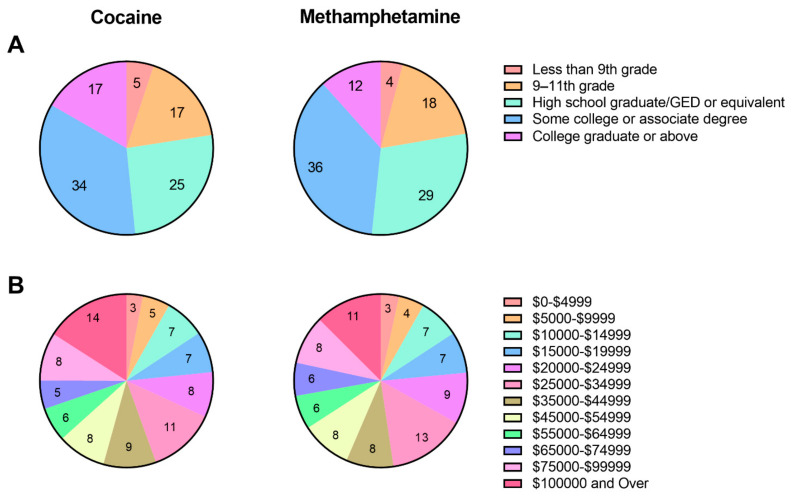
Demographic characteristics of people reporting lifetime use of cocaine and methamphetamine in the NHANES 2005–2018. **A** represents Education Level and **B** represents Annual Household Income. Numbers on each wedge represent percent of the sample with the characteristic of interest, rounded to the nearest whole number. GED = General Educational Development Test.

**Table 1 ijerph-18-12259-t001:** Estimated frequencies of past-month cocaine and methamphetamine use by survey years, NHANES 2005–2018. Frequencies were extrapolated to the U.S. population using weights provided by NHANES for each cycle between 2005–2018.

Survey Year	Cocaine—*n*	Methamphetamine—*n*
2005–2006	2,218,308	1,062,015
2007–2008	2,364,077	765,305
2009–2010	2,620,742	227,261
2011–2012	2,651,675	830,829
2013–2014	1,653,094	1,033,882
2015–2016	2,513,407	874,046
2017–2018	3,969,027	1,735,796

**Table 2 ijerph-18-12259-t002:** Demographic and other drug use characteristics of the sample reporting lifetime use of cocaine and methamphetamine in the NHANES 2005–2018.

	Cocaine (*n* = 3975)	Methamphetamine (*n* = 1606)
**Average age—years (±SD)**	40.33 (±11.41)	40.50 (±10.88)
**Sex—no. (%)**		
Male	2494 (62.7)	1026 (63.9)
Female	1481 (37.3)	580 (36.1)
**Average age of onset of use—years (±SD)**	21.62 (±5.77)	22.25 (±6.98)
**Other drug use—no. (%)**		
Regular tobacco use	1871 (47.1)	815 (50.7)
Regular cannabis use	1869 (47.0)	846 (52.7)
Lifetime heroin use	548 (13.8)	301 (18.7)
Lifetime cocaine use	3975 (100)	1469 (91.5)
Lifetime methamphetamine use	1469 (37.0)	1606 (100)
**Lifetime IV use—no. (%)**	472 (11.9)	332 (20.7)

Notes: IV = intravenous.

**Table 3 ijerph-18-12259-t003:** Hierarchical multiple regression results for age of onset of cocaine use.

	B	95% CI for B		se B	β	R^2^	ΔR^2^
		LL	UL				
**Block 1**						0.094	0.093 **
Constant	16.663	15.013	18.313	0.841			
Survey Cycle	−0.217	−0.392	−0.041	0.089	−0.054 *		
Age	0.155	0.132	0.177	0.011	0.304 **		
**Block 2**						0.156	0.151 **
Constant	18.327	16.253	20.401	1.057			
Survey Cycle	−0.119	−0.290	0.052	0.087	−0.030		
Age	0.159	0.137	0.181	0.011	0.312 **		
Sex (ref = male)	−0.239	−0.746	0.268	0.258	−0.020		
Education (ref = less than 9th grade)	−0.018	−0.269	0.233	0.128	−0.003		
Income (ref = $0 to $4999)	−0.117	−0.176	−0.057	0.030	−0.091 **		
Regular Tobacco Use (ref = no regular use)	0.889	0.364	1.415	0.268	0.076 **		
Regular Cannabis Use (ref = no regular use)	−1.159	−1.684	−0.634	0.268	−0.095 **		
Lifetime Heroin Use (ref = no lifetime use)	−0.543	−1.162	0.077	0.316	−0.041		
Lifetime Methamphetamine Use (ref = no lifetime use)	−1.540	−2.043	−1.038	0.256	−0.134 **		
Lifetime IV Use (ref = no lifetime use)	−1.386	−2.162	−0.610	0.396	−0.086 **		

Notes: B = unstandardized regression coefficient; CI = confidence interval; LL = lower limit; UL = upper limit; se B = standard error of the coefficient; β = standardized coefficient; R^2^ = coefficient of determination; ΔR^2^ = adjusted R^2^. * *p* < 0.05; ** *p* < 0.001.

**Table 4 ijerph-18-12259-t004:** Hierarchical multiple regression results for age of onset of methamphetamine use.

	B	95% CI for B		se B	β	R^2^	ΔR^2^
		LL	UL				
**Block 1**						0.089	0.087 ***
Constant	12.331	9.258	15.404	1.566			
Survey Cycle Year	0.263	−0.062	0.588	0.165	0.053		
Age	0.193	0.150	0.236	0.022	0.291 ***		
**Block 2**						0.147	0.136 ***
Constant	12.834	8.297	17.371	2.311			
Survey Cycle	0.341	0.022	0.659	0.162	0.068 *		
Age	0.215	0.172	0.258	0.022	0.323 ***		
Sex (ref = male)	−0.239	−0.746	0.268	0.258	−0.020		
Education (ref = less than 9th grade)	0.070	−0.418	0.559	0.249	0.010		
Income (ref = $0 to $4999)	−0.220	−0.333	−0.107	0.058	−0.133 ***		
Regular Tobacco Use (ref = no regular use)	1.919	0.950	2.888	0.494	0.132 ***		
Regular Cannabis Use (ref = no regular use)	−1.311	−2.263	−0.360	0.485	−0.090 **		
Lifetime Heroin Use (ref = no lifetime use)	1.124	−0.126	2.374	0.637	0.066		
Lifetime Cocaine Use (ref = no lifetime use)	0.835	−1.017	2.687	0.944	0.029		
Lifetime IV Use (ref = no lifetime use)	−1.614	−2.875	−0.353	0.642	−0.095 *		

Notes: B = unstandardized regression coefficient; CI = confidence interval; LL = lower limit; UL = upper limit; se B = standard error of the coefficient; β = standardized coefficient; R^2^ = coefficient of determination; ΔR^2^ = adjusted R^2^. * *p* < 0.05; ** *p* < 0.01; *** *p* < 0.001.

## Data Availability

The data in the present study were extracted from the public CDC NHANES database: https://wwwn.cdc.gov/nchs/nhanes/.

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
