# Peer review of "Age of Onset and Its Related Factors in Cocaine or Methamphetamine Use in Adults from the United States: Results from NHANES 2005–2018"

_ijerph, 2021, doi:10.3390/ijerph182212259_

Round 1

Reviewer 1 Report

I found several typos that should be corrected. Below some examples:

  • Line 29: 27 million people have used
  • Line 33: increasing
  • Line 33: harms are associated
  • Line 35: fivefold times
  • Line 38: reported
  • Line 52: psychostimulant use
  • Line 305: Results in the present paper suggests

Line 84 and 95: (Centers for Disease 84 Control and Prevention, 2021) should this be a reference?

2.1. Study population

The authors provided a citation regarding details about sampling and data collection. However, it would be helpful if the authors provide a brief description of the sampling method, which help the readers to evaluate the representativeness of the sample.

To secure representativeness national survey data are usually weighted to account for differences regarding particular population distributions such as gender, level of education, region etc. The present description lacks a note on weighting. Please proved details on the procedures used.

2.2.1. Drug use questionnaire:

Line 100: Regular tobacco use (over 100 100 cigarettes in lifetime). Depending on the respondent’s age I would question that over 100 cigarettes can be defined as regular use. For example, if the respondent’s age is 40 (average age provided in Tab 2), an assuming 18 as age of onset, this would mean an average consumption of 3.33 cigarettes per year. In fact, regular cannabis use is defined using a different frequency (1 or more times per month in a year). Furthermore, recalling the exact number of cigarettes in a lifetime is subject to severe recall biases. I don’t know this survey, but given that for cannabis past-year use was use, couldn’t be done the same with cigarettes?

Figure 1.

Since a long time period is analysed I think it would be more useful to graph prevalence instead of the frequencies, as I believe also the population changed a lot over this long timespan. But this is a suggestion.

Table 2.

The table title should be self-explanatory, i.e. the survey years of the pooled sample should be specified.

2.3. Statistical analyses and 3. Results

It does not seem that from the analysed data a clear growing trend is shown. For example, at the end of the day the prevalence of past-month cocaine use changed only from 1.178 in 2005-2006 to 1.820 in 2017-2018. In order to state that the prevalence of both measures has increased over time, it would be useful to include tests of linear trends, to test the null hypothesis of no change in prevalence over time.

As a suggestion, and linking this to my previous comment, if the authors will go for figures illustrating trends in prevalence, the figures could be useful to show percentage differences in past-month cocaine and methamphetamine use. Each point could denoted the percentage change compared to the previous year (e.g. for 2017-2018, the reference year could be 2015-2016).

As found by Kjellsson et al. (2014), the length of recall period in self-reported health care questions varies between surveys, and this variation may affect the results of the studies. Since the age at survey has a significant effect in the present analyses, I suggest that the authors evaluate the feasibility of performing a robustness check in which the analyses are re-run on a younger subsample, by setting the upper age limit at a younger age.

Lines 295-296 “In addition, results from the present study may reduce stigma towards people who use psychostimulants, methamphetamine in particular.” What stigma do the authors refer to? A reference would also be useful here.

Line 307 “In fact, considering age of initiation of cannabis use is lower than both cocaine and methamphetamine” it seems a strong statement, as this is not investigated in the paper. I would refer to this finding saying “has been found by previous studies..” or similar

Line 308 “it may be that cannabis use predates experimentation with psychostimulants.” This comment is really unclear to me and should be provided with proper references.

In the Discussion section the authors elaborate on the role of polysubstance use without defining it. If polysubstance use is intended following the common definition of a consumption of more than one drug at once, the authors did not analyse it, so everything concerning the topic should be presented as an hypothesis.

4.3. Limitations

I think that the biggest limitation of this study should be clearly stated here. Working with the age of onset is usually problematic due to recall biases. Here the problem is even more serious, as the authors do not set lower and upper age limits for their sample in order to minimise the potential issues of recall error with respect to age at first use. A statement concerning this limitation should be added.

5. Conclusions

In the Conclusions the authors state that “Taken together, these findings challenge the existing approach in the “War on Drugs”. Targeting supply rather than users may reduce prevalence. They did not make reference to this point in the discussion section. Although from a personal point of view I can agree with them and I also that the conclusions of a paper like this should provide some policy implications, this conclusion should be supported by some clearer reasoning in the discussion section or otherwise removed.

Linking to the previous remark, overall the conclusions might be better structured in order to provide clearer inputs for what the authors say it’s their aim, i.e. provide inputs for the development of new prevention programs to reduce psychostimulant use.

References:

Kjellsson, G., Clarke, P.M., & Gerdtham, U. (2014). Forgetting to remember or remembering to forget: a study of the recall period length in health care survey questions. Journal of health economics, 35, 34-46 .

Author Response

Dear Reviewer,

We would also like to thank you for your detailed and valuable feedback and insightful comments, which have substantially helped us to improve the manuscript.

Please find attached a revised version of the manuscript, in which we have addressed all the suggestions and comments. The response to comments are in italics and blue font. The changes in the manuscript file are tracked.

We hope the revisions are satisfactory and the manuscript is now suitable for publication in the International Journal of Environmental Research and Public Health.

Sincerely,

Dr Alexandre Guerin and Associate Professor Jee Hyun Kim

Review 1

I found several typos that should be corrected. Below some examples:

  • Line 29: 27 million people have used
  • Line 33: increasing
  • Line 33: harms areassociated
  • Line 35: fivefoldtimes
  • Line 38: reported
  • Line 52: psychostimulant use
  • Line 305: Results in the present paper suggests

We thank the reviewer for their thorough review. We have now corrected these typos (Lines 29, 33, 35, 52, 329).

Line 84 and 95: (Centers for Disease 84 Control and Prevention, 2021) should this be a reference?

Yes, this has now been added to the reference list.

2.1. Study population

The authors provided a citation regarding details about sampling and data collection. However, it would be helpful if the authors provide a brief description of the sampling method, which help the readers to evaluate the representativeness of the sample.

We have added the following information about sampling: “The sample for the survey is selected to represent the U.S. population of all ages. In addition, NHANES over-samples persons 60 and older, African Americans, and Hispanics to produce reliable statistics of potentially under-represented groups.” (Lines 87-89).

To secure representativeness national survey data are usually weighted to account for differences regarding particular population distributions such as gender, level of education, region etc. The present description lacks a note on weighting. Please proved details on the procedures used.

In addition to the original details on the weighting (now Lines 116-119), we have added the following information to further clarify NHANES creation of weights: “Sample weights are created in three steps; base weights for individuals are first computed, then adjusted for non-response, and finally post-stratified to match the population control totals for each sampling subdomain.” (Lines 119-122).

2.2.1. Drug use questionnaire:

Line 100: Regular tobacco use (over 100 100 cigarettes in lifetime). Depending on the respondent’s age I would question that over 100 cigarettes can be defined as regular use. For example, if the respondent’s age is 40 (average age provided in Tab 2), an assuming 18 as age of onset, this would mean an average consumption of 3.33 cigarettes per year. In fact, regular cannabis use is defined using a different frequency (1 or more times per month in a year). Furthermore, recalling the exact number of cigarettes in a lifetime is subject to severe recall biases. I don’t know this survey, but given that for cannabis past-year use was use, couldn’t be done the same with cigarettes?

We thank the reviewers for bringing this up to our attention. We have now changed the regular tobacco use variable to represent participants’ reporting of cigarettes use most days over the past month. This has been updated in the method section (Lines 102-103). We have also updated the regressions in Table 3 and 4 accordingly, as well as drug use characteristics in Table 2. Following this tobacco use update, lifetime heroin use was no longer associated with age of onset of cocaine use (Table 3). This association was weak to begin with before the tobacco use update (β = -0.045; p = 0.036), hence we believe the results are now more accurate thanks to the reviewer’s suggestion.

Figure 1.

Since a long time period is analysed I think it would be more useful to graph prevalence instead of the frequencies, as I believe also the population changed a lot over this long timespan. But this is a suggestion.

We agree with the reviewer and have replaced Figure 1 with a graph of prevalence over time. Estimated frequencies are now presented in Table 1 instead.

Table 2.

The table title should be self-explanatory, i.e. the survey years of the pooled sample should be specified.

We have updated the title of Table 2 to “Demographic and other drug use characteristics of the sample reporting lifetime use of cocaine and methamphetamine in the NHANES 2005-2018”

2.3. Statistical analyses and 3. Results

It does not seem that from the analysed data a clear growing trend is shown. For example, at the end of the day the prevalence of past-month cocaine use changed only from 1.178 in 2005-2006 to 1.820 in 2017-2018. In order to state that the prevalence of both measures has increased over time, it would be useful to include tests of linear trends, to test the null hypothesis of no change in prevalence over time.

As a suggestion, and linking this to my previous comment, if the authors will go for figures illustrating trends in prevalence, the figures could be useful to show percentage differences in past-month cocaine and methamphetamine use. Each point could denoted the percentage change compared to the previous year (e.g. for 2017-2018, the reference year could be 2015-2016).

We have now tested the linearity of trends for prevalence, frequencies, and age of onset of cocaine and methamphetamine use. The Statistical analyses section was updated accordingly: “In addition, linearity of trends was tested using simple linear regressions” (Lines 130-131). We also updated the result section accordingly: “While past-month use of both substances in the U.S. adult population has reached a new maximum in prevalence (Figure 1) and frequency (Table 1) in recent years, these changes were not statistically significant when tested for linear trends (ps > 0.05).” (Lines 145-147).

We have now added percentage change for each survey year on Figure 1, and discussed it in the result section: “Notably, there was a marked increase in prevalence between the last two survey cycles, with a percent change increase of 57.1% and 97.5% from 2015-2016 to 2017-2018 for cocaine and methamphetamine, respectively” (Lines 151-154). In addition, the following was added to the statistical analysis section: “To examine change between each survey year, the percent change in prevalence was calculated for each cycle” (Lines 124-125).

As found by Kjellsson et al. (2014), the length of recall period in self-reported health care questions varies between surveys, and this variation may affect the results of the studies. Since the age at survey has a significant effect in the present analyses, I suggest that the authors evaluate the feasibility of performing a robustness check in which the analyses are re-run on a younger subsample, by setting the upper age limit at a younger age.

We agree with the reviewer that age may be a confounding factor. In the original manuscript, age was included as a control variable in both regressions (now Lines 133-134). Even with significant age effects, addition of the variables of interest significantly improved the two models. Keeping in mind that the sample size is limited in these analyses were within people reporting to use cocaine and/or methamphetamine, we believe controlling for age in the regressions is the best approach, rather than to further limit the sample sizes by stratifying by age groups. We do understand it is a limitation, and have added the following statement in the Limitations section, and cited Kjellsson et al. (2014): “Evidence suggests that the length of recall period in self-reported health care questions varies between survey questions [45], and may affect the results of the survey. In the present study, participant age ranged from 18 to 69 years-old. The age of onset measure in particular may have been subject of recall bias, as older participant may not correctly recall their exact age at first use. While we did control for the age at survey in our regressions, future studies should aim to reproduce these findings in a younger population with a narrower age range.” (Lines 360-366)

Lines 295-296 “In addition, results from the present study may reduce stigma towards people who use psychostimulants, methamphetamine in particular.” What stigma do the authors refer to? A reference would also be useful here.

We have added the following sentence in the revised manuscript: “People who use substances experience high level of stigma such as stereotyping (i.e., being perceived as dangerous, immoral, and having a lower socioeconomic status), discrimination and eliciting negative emotional reactions [35], which may hinder access to treatment and care [36,37].” (Lines 316-319)

Line 307 “In fact, considering age of initiation of cannabis use is lower than both cocaine and methamphetamine” it seems a strong statement, as this is not investigated in the paper. I would refer to this finding saying “has been found by previous studies..” or similar

Thank you for the suggestion. We have now rephrased this statement to: “Considering that other studies have found that the average age of initiation of cannabis use is lower than cocaine or methamphetamine use […]” (Lines 330-331)

Line 308 “it may be that cannabis use predates experimentation with psychostimulants.” This comment is really unclear to me and should be provided with proper references.

We have now referenced this statement (Line 332).

In the Discussion section the authors elaborate on the role of polysubstance use without defining it. If polysubstance use is intended following the common definition of a consumption of more than one drug at once, the authors did not analyse it, so everything concerning the topic should be presented as an hypothesis.

This is a very important point, and we thank the reviewer for highlighting it. We have changed the term polysubstance use to “lifetime use of multiple substances” throughout the discussion as we cannot ascertain whether substances were consumed at the same time, and therefore polysubstance was not assessed in this study.

4.3. Limitations

I think that the biggest limitation of this study should be clearly stated here. Working with the age of onset is usually problematic due to recall biases. Here the problem is even more serious, as the authors do not set lower and upper age limits for their sample in order to minimise the potential issues of recall error with respect to age at first use. A statement concerning this limitation should be added.

We have noted in the discussion that age of onset may have been overreported in the original manuscript: “It should also be noted that the results from this study is likely an overestimation of the actual age of onset, as all participants were aged 18 and above.” (now Lines 301-303). In addition, we have now added the following statement in the limitation section: “Evidence suggests that the length of recall period in self-reported health care questions varies between survey questions [45], and may affect the results of the survey. In the present study, participant age ranged from 18 to 69 years-old. The age of onset measure in particular may have been subject of recall bias, as older participant may not correctly recall their exact age at first use. While we did control for the age at survey in our regressions, future studies should aim to reproduce these findings in a younger population with a narrower age range.” (Lines 360-366)

Conclusions

In the Conclusions the authors state that “Taken together, these findings challenge the existing approach in the “War on Drugs”. Targeting supply rather than users may reduce prevalence. They did not make reference to this point in the discussion section. Although from a personal point of view I can agree with them and I also that the conclusions of a paper like this should provide some policy implications, this conclusion should be supported by some clearer reasoning in the discussion section or otherwise removed.

We have added the following statement in the discussion: “Importantly, the increase in cocaine and methamphetamine use appears to be largely supply-driven. Findings from this study challenge the idea of the “War on Drugs” [25] that heavily involves incarceration of drug users. The present findings highlight the need for law enforcements to target suppliers and manufacturers, rather than users.” (Lines 287-291)

Linking to the previous remark, overall the conclusions might be better structured in order to provide clearer inputs for what the authors say it’s their aim, i.e. provide inputs for the development of new prevention programs to reduce psychostimulant use.

We have now revised the conclusion and added the following: “Considering the strong association between early onset of use and higher household income, education programs should target schools of higher sociodemographic status to prevent early psychostimulant use. In addition, this study suggests that the use of one drug early in life may lead to the use of other substances later. Promoting the treatment of people who started using drugs early with financial aids and subsidies may therefore presents a unique opportunity to prevent their transition to other drug use and dependence.” (Lines 379-385) 

References:

Kjellsson, G., Clarke, P.M., & Gerdtham, U. (2014). Forgetting to remember or remembering to forget: a study of the recall period length in health care survey questions. Journal of health economics, 35, 34-46 .

Reviewer 2 Report

Dear authors

The manuscript is well written, clear cristal and concise.

The references are actual and support the manuscript. 

There are minor things to be corrected, namely the use of abbreviations, once we use them for the first time from that moment we should only use the abbreviations. 

Regarding tables, usually, scientific tables don't have vertical lines, and the legend of abbreviations is not in the title but below in the legend of abbreviations.

Applicable to all tables. 

In order to facilitate the results reading, a suggestion authors could use more figures, in pie or bars.

Very interesting article and the conclusions are supported by the obtained results.

Author Response

Dear Reviewer,

We would also like to thank you for your detailed and valuable feedback and insightful comments, which have substantially helped us to improve the manuscript.

Please find attached a revised version of the manuscript, in which we have addressed all the suggestions and comments. The response to comments are in italics and blue font. The changes in the manuscript file are tracked.

We hope the revisions are satisfactory and the manuscript is now suitable for publication in the International Journal of Environmental Research and Public Health.

Sincerely,

Dr Alexandre Guerin and Associate Professor Jee Hyun Kim

Reviewer 2

The manuscript is well written, clear cristal and concise.

The references are actual and support the manuscript. 

We greatly appreciate the reviewer for reviewing our manuscript and the positive feedback.

There are minor things to be corrected, namely the use of abbreviations, once we use them for the first time from that moment we should only use the abbreviations. 

We thank the reviewer for bringing this to our attention, which we now have corrected (e.g. Line 83). Notably, we have redefined every acronym in Figure and Table captions according to APA guidelines.

Regarding tables, usually, scientific tables don't have vertical lines, and the legend of abbreviations is not in the title but below in the legend of abbreviations.

Applicable to all tables. 

We have removed vertical lines from all tables, and have moved the legend of abbreviations to the bottom of each table. We’d like to ensure the reviewer that if accepted for publication, the journal typesetter will further advise and guide the table formatting. 

In order to facilitate the results reading, a suggestion authors could use more figures, in pie or bars.

We would like to thank the reviewer for the suggestion. We have now removed Education and Annual Household Income from Table 2 and have graphed these characteristics as pie-charts, displayed in Figure 3.

Very interesting article and the conclusions are supported by the obtained results.

Thank you for positive feedback.

Round 2

Reviewer 1 Report

Thanks to the Authors for implementing my suggestions, I believe they did a great job.

Concerning the conclusions, I still believe that the support that the  manuscript's results provide for challenging the existing approach in the "War on Drugs" and therefore for targeting supply rather than users may reduce prevalence is limited, although I totally agree with it. To this purpose it would have been useful to include some measure of reported perceived availability of the drug, although I can understand this is not possible because not investigated in the survey. Overall I do find the paper interesting for the reader and I recommend its publication.